# The Global Demand and Supply Balance of the Human Papillomavirus Vaccine: Implications for the Global Strategy for the Elimination of Cervical Cancer

**DOI:** 10.3390/vaccines12010004

**Published:** 2023-12-19

**Authors:** Stefano Malvolti, Adam Soble, Paul Bloem, D Scott LaMontagne, Rakesh Aggarwal, Punnee Pitisuttithum, Helen Rees, Tania Cernuschi

**Affiliations:** 1MMGH Consulting GmbH, 8049 Zurich, Switzerland; soblea@mmglobalhealth.org; 2Department of Immunization, Vaccines and Biologicals, World Health Organization, 1202 Geneva, Switzerland; bloemp@who.int (P.B.); cernuschit@who.int (T.C.); 3John Snow, Inc. (JSI), Arlington, VA 22202, USA; scott_lamontagne@jsi.com; 4Jawaharlal Institute of Postgraduate Medical Education and Research, Puducherry 605006, India; aggarwal.ra@gmail.com; 5Faculty of Tropical Medicine, Mahidol University, Bangkok 10400, Thailand; punnee.pit@mahidol.ac.th; 6Wits RHI, University of Witwatersrand, Johannesburg 2001, South Africa; hrees@wrhi.ac.za

**Keywords:** human papilloma virus, vaccine, demand forecast, supply, cervical cancer

## Abstract

As of November 2023, 140 World Health Organization (WHO) member states had introduced human papillomavirus (HPV) vaccination in their routine immunization schedules. Despite a continuously increasing demand from countries across all income groups, supply constraints, COVID-19 pandemic disruptions, and other factors have slowed the pace of introduction, particularly in low-resource settings. Using a population-based forecasting methodology and leveraging the WHO’s yearly vaccine supply data collection, we updated global demand and supply projections for the HPV vaccine for the period of 2022–2031. The analysis aimed at clarifying the magnitude of the challenges to bringing in equitable access to HPV vaccines, which can hinder the achievement of the Global Strategy for the Elimination of Cervical Cancer. The results of this analysis show that the risk of HPV shortages has significantly decreased, and global supply is now, under normal circumstances, sufficient to meet global demand. In the long term, HPV supply will be more than sufficient to meet the Global Strategy’s goal of 90% of girls fully vaccinated with the HPV vaccine by the age of 15 years. Nonetheless, paying attention to the formulation of policies and carefully managing demand and supply will be required to ensure the long-term sustainability of the HPV vaccine program.

## 1. Introduction

Human papillomavirus (HPV) is a common virus that infects the anogenital regions and the reproductive tract. Persistent infection with high-risk HPV subtypes is strongly associated with the development of cervical and other cancers [1]. Vaccines to prevent HPV-related disease were first licensed in 2006 [2], and recognition of their importance has grown steadily in the past 16 years. In 2009, the World Health Organization (WHO) recommended the inclusion of HPV vaccines in national immunization programs, prioritizing the prevention of cervical cancer by vaccinating girls before sexual debut [3]. To enable more equitable access to HPV vaccines, Gavi, the Vaccine Alliance, the main financing mechanism for low- and lower-middle-income countries, has been supporting HPV vaccination since 2012. In 2020, the World Health Assembly endorsed the Global Strategy for Cervical Cancer Elimination [4], defining a vision of a world where cervical cancer is eliminated as a public health problem. A threshold of 4 cases per 100,000 women each year was set as the target. To achieve this goal, the 90-70-90 by 2030 target was defined for countries to be on the path towards elimination:-90% of girls fully vaccinated with the HPV vaccine by the age of 15 years.-70% of women screened with a high-performance test by 35 years of age and again by 45 years of age.-90% of women identified as having cervical disease receiving treatment (90% of women with precancer treated, and 90% of women with invasive cancer managed).

Due to high interest in preventing HPV-related diseases through vaccination, and also the time needed to increase manufacturing capacity, the global supply of HPV vaccines has been insufficient to meet global demand since 2014–2015. The WHO recommendation of performing large multi-age cohort (MAC) campaigns in the first year of introduction with the goal of accelerating the reduction in the cervical cancer burden further increased the pressure on supply [5]. The constraints led to the adjustment of introduction plans, especially in Gavi-supported countries. To address this supply situation and maximize the public health impact of available HPV vaccines, the WHO’s Strategic Advisory Group of Experts on Immunization (SAGE) recommended in 2019 that countries pause HPV vaccination in secondary target groups, such as boys and older girls, in order to relieve supply constraints and instead focus on enabling access for the highest-priority populations in high-burden countries [6]. Based on the latest scientific evidence on HPV dose requirements and their public health implications, in 2022, SAGE recommended updating vaccination schedules to use one- and two-dose regimens for younger and older populations, respectively, based on accumulated evidence that reduced-dose schedules provide efficacy comparable to the current two- and three-dose regimens [7]. 

This article, drawing from the work of the WHO’s Market Information for Access (MI4A) initiative [8], provides a current, comprehensive view of the dynamics of global HPV vaccine demand and supply, and tries to forecast market evolution in the next decade. It is intended to inform policy and implementation decisions about HPV vaccination, with the goal of enabling greater and more equitable access to HPV vaccines across all countries, regardless of region or income level.

## 2. Methods

### 2.1. Demand Forecasts

An unconstrained global forecast of the programmatic dose requirements (PDRs) for the HPV vaccine was developed for the 10-year period of 2022–2031 based on historical procurement data, the latest available information on country introduction plans, and key drivers of demand such as future vaccination coverage trends and timing of multi-age cohort campaigns (MACs). Programmatic dose requirements: In future years, the average estimated number of doses a country will need to procure to meet its immunization program needs, whether these are routine—national or subnational—campaigns/supplementary immunization activities or for special risk groups only. This requirement includes wastage (depending on the presentation) and buffer. 

Historical procurement data were obtained from the WHO/United Nations Children’s Fund (UNICEF) Joint Reporting Process [9] and from the UNICEF Supply Division. Planned and projected country introductions as well as MAC timing were obtained through the Joint Reporting Process, from Gavi operational forecasts, and from the WHO’s HPV program. Historical HPV coverage data were obtained from the 2021 WHO/UNICEF Estimates [10]. The impact of vaccine hesitancy was assumed to be constant, as captured in the historical coverage data. Population forecasts for the relevant age groups were obtained from the 2019 Revision of World Population Prospects [11]. 

Different demand scenarios were developed to test the impact of alternative policy and strategy decisions (assumptions are shown in Table 1). Scenarios were not supply constrained, included both public and private sector demand, and assumed that programmatic delays or disruptions due to COVID-19 will be fully resolved by the end of 2023. They assumed that MACs are implemented only for vaccination of girls (10 to 14 years of age) and that all countries will introduce the HPV vaccine by 2030, in accordance with the Global Strategy for Cervical Cancer Elimination. Following the introduction of the HPV vaccine, specific coverage uptake assumptions are applied based on historical coverage trends in the first four years of routine vaccine use (Year 1: 100% of the reference coverage, Y2: 80%, Y3: 90%, Y4: 100%). Starting from the fifth year of routine HPV vaccine use, coverage is assumed to increase by 0.5% per year until countries reach 90% coverage. The base case scenario reflects WHO recommendations as of 2019. Scenarios A–D reflect policy alternatives and do not represent WHO endorsement of specific target populations or dosing schedules. 

Additional details on HPV vaccine demand forecasting are provided in the Appendix A.

### 2.2. Supply Forecasts

Data on the available supply for commercialization (ASC) were accessed from the annual survey of vaccine manufacturers conducted by the WHO MI4A initiative and through discussions with manufacturers on future capacity prospects. Available supply for commercialization: the number of doses available for sale at the global level in one typical year with normal production facility utilization across the various vaccines (not factoring in special market, regulatory, or technical events). This differs from the manufacturing capacity or the plant yearly throughput. To manage the commercial sensitivity and risks of anti-trust violations stemming from those data, they are shown only in aggregate. Additional data were collected through review of publicly available clinical trial information, manufacturing process documentation, and vaccine product registration information. 

Data from all the sources combined were used to forecast the supply of the HPV vaccine for the next 10 years under two scenarios: a base case and a low-supply scenario. The base case supply scenario assumed that existing suppliers will remain committed to manufacturing HPV vaccines and will expand capacity. The base case scenario also assumed that the manufacturers currently developing new HPV vaccines will successfully complete their clinical development, obtain marketing authorization and WHO Prequalification, and make the expected quantities of supply available to all countries. The low-supply scenario used more conservative assumptions for increases in manufacturing capacity, where the production scale-up is reduced in size and some products do not successfully complete clinical development.

### 2.3. Supply/Demand Balance

Supply/demand balance was assessed by calculating the demand-to-supply ratio using each combination of demand and supply scenarios. In the absence of clear information on the evolution of country product preferences, timing of future availability of data supporting vaccination of boys, and applicability of a single-dose schedule for individual vaccines, as well as on potential procurement strategies, all vaccines were considered as contributing to the global supply in all scenarios.

Calculations were performed for short-term (1–3 years), medium-term (4–6 years), and long-term (7–9 years) time periods. To respect data confidentiality, results were coded as follows:Excess supply: ASC ≥ 2 × Demand.No risk of shortages: 1.3 × Demand ≤ ASC < 2 × Demand.Some risk of shortages: 1.1 × Demand ≤ ASC < 1.3 × Demand.Insufficient supply: ASC < 1.1 × Demand.

### 2.4. Price

Vaccine prices were accessed as reported by countries to the MI4A initiative [12] and supplemented with publicly disclosed prices through the Pan American Health Organization Revolving Fund (PAHO RF) [13] and the U.S. Centers for Disease Control and Prevention [14]. 

## 3. Results

### 3.1. Global Demand

Globally, 140 WHO member states have introduced HPV vaccination in their routine immunization schedules as of November 2023. Together, these countries are home to approximately forty percent of the target population of girls under 15 years old. When expressed in terms of world population coverage (i.e., weighted by population size), global coverage of the final HPV dose is estimated at 12% as of the end of 2021 [10]. Eighteen additional countries were forecasted to introduce the HPV vaccine during 2023, and of those, three introduced it at the end of 2022, ten introduced it in 2023, and five have still not introduced it [15]. Of the countries that have introduced the HPV vaccine, 47% self-procure and the remainder use one of the pooled procurement mechanisms (i.e., UNICEF or the PAHO Revolving Fund). A greater proportion of high-income countries (HICs) have introduced HPV vaccines than low- and middle-income countries that are eligible for Gavi support (Figure 1). Middle-income countries (MICs) that are not eligible for Gavi support or those that do not procure vaccines through the PAHO RF also lag in HPV vaccine introduction.

Global HPV programmatic dose requirements (PDR) in the base case scenario are estimated to have reached approximately 80 million doses in 2022 (Figure 2). This estimate incorporates the effects of the active PDR management performed by public health agencies (WHO and Gavi) in the past years in response to supply shortages as well as delays in program implementation due to the coronavirus disease (COVID-19) pandemic and other programmatic factors. Approximately 10% of the total PDR is estimated to be being used for the vaccination of boys in high-income country settings.

In the absence of supply constraints, PDR is expected to have risen sharply to over 130 million doses by 2023, with fluctuations between 120 and 140 million doses each year through to 2030 as additional countries introduce HPV vaccines and conduct MAC campaigns. PDR is projected to stabilize at approximately 125 million doses by 2031 once introductions and MAC campaigns are complete. The short-term increases in PDR are driven largely by the implementation of pending MAC campaigns in LICs and LMICs supported by Gavi and catch-up vaccination in selected large HICs. In the short-to-medium term, national introductions in China and India are expected to drive the most significant increases in global PDR; these two countries together will represent approximately one-third of the forecasted demand by 2031. The potential impact of a significant increase in vaccine hesitancy in the future is not captured in this analysis. 

Of the alternative scenarios modeled, Scenario A (one-dose schedule) gives the lowest dose requirements, with PDR stabilizing at around 70 million doses per year. Scenario B, which combines a switch to a one-dose schedule with gender-neutral vaccination, results in PDR estimates similar to the base case scenario in the medium and long terms. Scenario C, which adds gender-neutral vaccination in HICs and MICs to the base case scenario (two-dose schedule), yields higher PDR estimates, especially in the short and medium terms, with a peak PDR of 162 million doses in 2026. Finally, Scenario D, the PDR to support WHO’s cervical cancer elimination goals by 2030, gives the highest PDR in the long term due to increases in HPV coverage required in all countries, with the need for approximately 150 million doses per year by 2030. 

### 3.2. Global Supply

As of September 2023, six HPV vaccines are licensed and seven more are in Phase 3 clinical trials (Table 2), with pipeline vaccines trending towards higher valency. The majority of doses are manufactured by Merck Sharp & Dohme (MSD) for its two vaccines. MSD has also licensed three other companies to produce its four-valent vaccine: Indonesia’s PT Biofarma, Brazil’s Instituto Butantan, and Argentina’s Sinergium Biotech. Other countries are served through distribution agreements. In 2021, a bivalent HPV vaccine produced by Xiamen Innovax Biotech (Innovax) was prequalified by the WHO, expanding the prequalified supplier base to three companies. WHO prequalification is necessary to access UN procurement (UNICEF and PAHO), a tool used by more than 50% of the countries to purchase vaccines. All current suppliers are targeting, or are expected to target, the global market through WHO prequalification and marketing authorization in multiple markets. 

Technology transfers are limited for the HPV vaccine. MSD currently supplies a bulk HPV vaccine to its licensees, which then fill and finish the final product. Other technology transfers have been under discussion with manufacturers in Russia and Thailand; how these transfers will affect global HPV vaccine supply is unclear and has not been considered in this analysis.

Due to investments to increase manufacturing capacity and the entry of new manufacturers, supply has increased on average by 15% annually in recent years. Significant further increases are anticipated in 2023–2025. The base case projection foresees a three-fold increase in available supply over the medium term (4–6 years, range 2–4×), from the ~80 million doses expected to be available in 2022 (Figure 3). 

### 3.3. Supply/Demand Balance

The HPV supply/demand balance has improved in recent years due to increases in HPV vaccine supply, proactive management of HPV demand, and, unfortunately, as a result of the declines in demand consequent to the reductions in immunization coverage due to the COVID-19 pandemic. 

As result of those efforts, starting from 2022, global supplies are expected to meet base case global demand (Figure 4); however, risks of shortages remain at the country level, given the limited buffer supply and product preferences on the part of countries. Furthermore, large catch-up campaigns targeting older cohorts and the widespread adoption of gender-neutral vaccination strategies could still cause global shortages that could result in delayed access for other countries. Careful phasing of MAC campaigns and promoting country flexibility to use any of the HPV vaccines available will be needed in the next 2–3 years to enable all countries to achieve their HPV vaccination goals. 

The supply/demand balance is expected to improve steadily, such that supply will exceed base case demand from 2024 onward even in the low-supply scenario. Subject to vaccines being made available to all countries, this condition will enable countries to freely choose products with higher valency and implement vaccination strategies with less concern about adversely impacting the vaccine supply available to other countries.

Widespread adoption of one-dose schedules would shift the supply/demand balance further, thus fully eliminating shortage risks and allowing more widespread implementation, an increasing number of cohorts to be included in MAC campaigns, and greater flexibility in product choice (Scenario A). In the mid-to-long term, because one-dose schedules could lead to lower steady-state demand, they could also impact the sustainability of the HPV vaccine market for manufacturers, who may respond with price increases or market exits due to decreased demand. Transition to one-dose schedules would require not only evidence supporting the single-dose efficacy of individual products but also careful management of supply and demand. 

Widespread implementation of one-dose schedules would also create supply and budgetary space for a more generalized adoption of gender-neutral vaccination strategies and immunization of older cohorts, particularly if countries from all income groups implement a one-dose schedule in the short term (Scenario B). In this scenario, vaccine demand will remain similar to the base case scenario, sustained by the efforts to reach additional target populations. On the other hand, the immediate implementation of gender-neutral vaccination in HICs and MICs with continuation of two-dose schedules (Scenario C) will create significant challenges for maintaining the available supply in the short term. This highlights the importance of carefully managing the timing of policy implementation and synchronizing expansion of the target population with supply availability to avoid negatively impacting the primary target population. Finally, in the elimination scenario (Scenario D), supplies can be maintained throughout the entire period, in view of the elimination of the MACs. 

### 3.4. Price

The reported price per dose of the HPV vaccine varies based on the procurement method, vaccine subtype, and country income group (Figure 5). While there are overlaps in HPV price between these country income groups, a clear tiered structure to global HPV pricing is in place, with the lowest prices paid by the UNICEF Supply Division (SD) and in PAHO RF procurement, where both pay less than USD 10 per dose. 

The median price for HPV2 and HPV4 products paid by self-procuring upper-middle-income countries (UMICs) is significantly higher than this level. This difference may influence national HPV introduction decision-making due to affordability barriers in countries without access to pooled vaccine procurement mechanisms. Generally, prices for the bivalent vaccine are lower, ranging from USD 10.25 to USD 14.14, compared with prices for the quadrivalent, which range from USD 13.18 to USD 64.16 for self-procuring UMICs, which reflect important differences in the price of HPV vaccines depending on the valency of a product.

Over the last 5 years, HPV vaccine prices have generally declined across all income and procurement groups. Further price reductions are expected as new market entrants (including the recent WHO prequalified HPV2 vaccine from Xiamen Innovax Biotech) increase competition, provided that demand is generated and sustained for those new products. 

## 4. Discussion

The HPV vaccine market is finally evolving towards a healthier supply/demand balance, which will support an acceleration of the health impact delivered by those vaccines. This condition comes after several years of severe supply constraints and mismatch between public health needs and supply allocation, driven, among the other factors, by the expansion of HPV vaccination programs for boys and older-age cohorts in higher-income countries ahead of the introduction in the lower-income countries with high burdens of the disease. This healthier balance has been the result of concerted actions in the management of HPV vaccination demand and supply. Demand has been proactively managed through adjustments to global policies and recommendations aimed at reducing the use of the HPV vaccine outside the primary target population, as well as through the optimized phasing of new introductions in LMICs. Efforts to improve the supply of HPV vaccines have included an open dialogue with manufacturers that has led to significant decisions in term of capacity increases, acceleration of submission for WHO prequalification, and reconfirmation of the commitment to supply HPV vaccines. 

Such dynamics are not uncommon in the vaccine ecosystem when the introduction of new vaccines is coupled with scarce supply: parallels can be seen between the historical access issues for HPV vaccines and the more recent challenges faced for equitable access to inactivated polio vaccines, malaria, typhoid conjugate vaccines, and, most recently, COVID-19 vaccines for countries across all income levels. Individual countries may legitimately prioritize the health impact for their own populations, and manufacturers may prioritize the available vaccine supply for high-price markets. These decisions often result in the suboptimal allocation of available vaccine supply and a reduced public health impact from a global perspective. 

When global health institutions play a brokering role between different interests, with a priority focus on global public health goals, more equitable access to vaccines and a greater health impact can be achieved, as happened for HPV vaccines starting in 2019. This active role must continue in the future; as supply constraints are fully relieved in the next 2–3 years, several issues will still require careful attention. 

Firstly, careful planning and management by countries, regional and global immunization agencies, and manufacturers will be required to manage the available supply and effectively support HPV vaccine introductions and MACs, especially in large countries, mindful that two-thirds of girls globally are still unreached.

Secondly, national immunization technical advisory groups (NITAGs) in countries across all income groups will need to take decisions that will have a fundamental impact on the evolution of HPV vaccine demand in the short term. Moving forward, opportunities to increase and facilitate the broadest possible access to HPV vaccines for all target populations should be pursued by the WHO and NITAGs. This should be done through timely revision of policy recommendations to incorporate emerging evidence, especially evidence on how products compare in terms of immunogenicity, efficacy, and effectiveness, and evidence for the optimization of the schedules in relation to the adoption of single-dose schedules for routine use for girls, boys, and in MACs. 

Thirdly, it will be necessary to monitor and inform the evolution of country product preferences, taking into account the impact of vaccine nationalism and diplomacy. Preferences that are too rigid can delay new vaccine introductions, exacerbate risks of supply shortages, and keep prices relatively and unnecessarily high, should few of the available HPV vaccines be desired by countries. According to the WHO, “Current evidence suggests that from the public health perspective the bivalent, quadrivalent and nonavalent vaccines offer comparable immunogenicity, efficacy and effectiveness for the prevention of cervical cancer, which is mainly caused by HPV types 16 and 18” [3]. To support flexible country product preferences and enable balance between supply and demand in the short term, it is necessary to strengthen information supplied to countries for evidence-based holistic decision-making on choices of HPV vaccine products. In the medium-to-long term, the arrival of second-generation, multivalent HPV vaccines may result in many countries switching to new, higher-valency HPV vaccines. 

Furthermore, vaccine affordability remains an important concern that needs continued attention and action. The number of introductions in LICs and MICs, which carry a disproportionate share of the HPV disease burden [16], remains lower than in HICs. This is particularly true for MICs that are not eligible for Gavi support or support from the PAHO RF, in which procurement and implementation costs can form a substantial fraction of health and immunization budgets. While price commitments to countries transitioning away from eligibility for Gavi support have been established, some countries are no longer eligible for these time-limited commitments [17]. This strain has increased in the context of the COVID-19 pandemic, current global economic uncertainties, and the Immunization Agenda 2030 (IA2030), which aims to regularly reach zero-dose children. More streamlined pricing can improve affordability as well as support more competitive procurement practices, as suggested by the overlap between prices that MICs report paying and those reported by HICs. Support is required, in particular, for self-procuring MICs, consistent with the potential assistance from Gavi to support the introduction of the HPV vaccine in MICs, as part of its 2021–2025 strategy [18]. 

Lastly, in the second half of the decade, supply is likely to exceed demand with a magnitude that may vary depending on the demand scenario. Such a condition will lead, most likely, to the exit of those suppliers that will see very limited demand for their product or that will see profit margins for their vaccines greatly reduced with the progressive reduction in price as result of the growing competition. As happened in the past for other vaccines [19]—most recently the Pentavalent vaccine [20]—this may lead to the re-creation of undesired oligopolistic or monopolistic market conditions, reduction in the innovation, and, in some extreme cases, to the return of supply constraints. To avoid such outcomes, ongoing dialogue will be required with manufacturers and large funding and procurement actors to ensure global demand evolution visibility and to explore the appropriate incentive systems for a diverse, sustainable manufacturing base, which is maintained to enable continued access and innovation. 

## 5. Conclusions

In the short, medium, and long terms, HPV vaccine markets will continue to evolve as result of the entry of new manufacturers and second-generation products; vaccine implementation in additional countries; and policy changes, including shifts to one-dose schedules and possibly further expansion of the target populations to boys and to older age groups. To achieve equitable access to supply, and to allow an active and dynamic management of demand and supply, a continued global-level access dialogue on access will be required among all stakeholders and the exploration of appropriate incentives to ensure that constraints and opportunities of all parties are identified well in advance, discussed, and serve as a basis for the engineering of interventions towards greater access and continued innovation.

## Figures and Tables

**Figure 1 vaccines-12-00004-f001:**
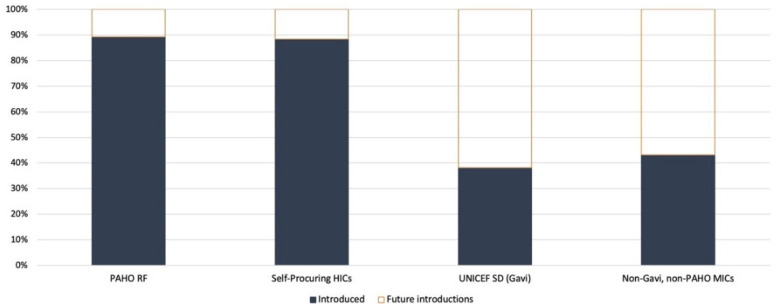
HPV vaccine introduction status by country procurement group in 2021.

**Figure 2 vaccines-12-00004-f002:**
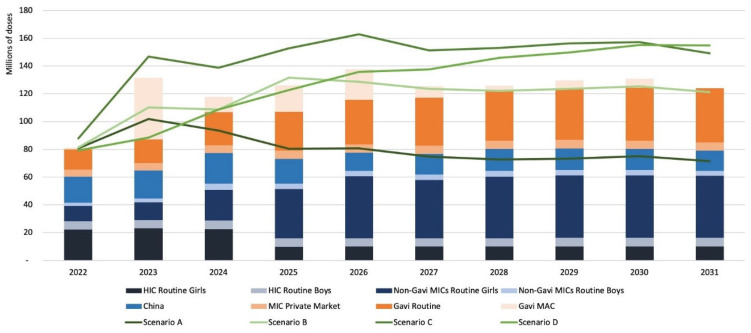
Comparison of unconstrained demand scenarios.

**Figure 3 vaccines-12-00004-f003:**
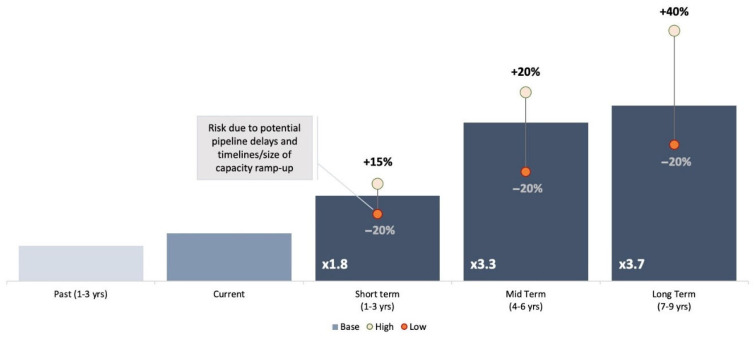
Available supply for commercialization forecast.

**Figure 4 vaccines-12-00004-f004:**
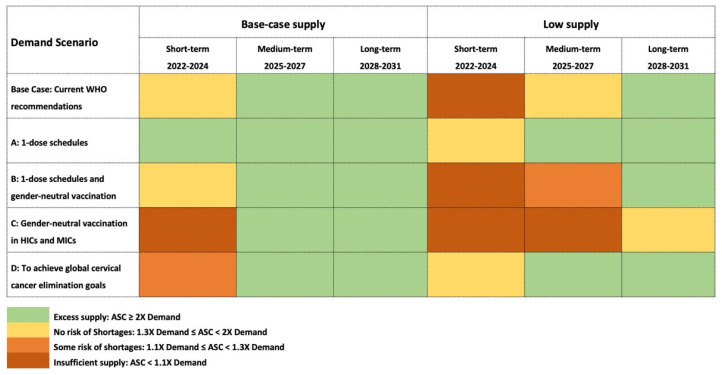
Supply/demand balance.

**Figure 5 vaccines-12-00004-f005:**
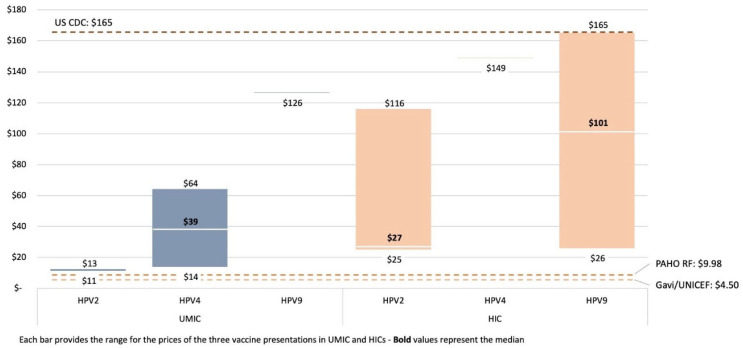
HPV self-procured price per dose (2021).

**Table 1 vaccines-12-00004-t001:** Demand forecasting scenarios.

Scenario	Primary Target Population	Secondary Target Populations
Base Case: Current WHO recommendations	Two-dose schedule for girls under 15 years old. All countries introduce by 2028.	Only currently planned MAC campaigns with two doses each are for girls under 15 years old and gender-neutral vaccination in countries with such existing recommendations (only girls in other countries).
Scenario A: One-dose schedules	One-dose schedule for girls under 15 years old. LICs and LMICs adopt the one-dose schedule in 2023, while UMICs and HICs (including China) have a delayed switch in 2025.	As in base case, except that MAC campaigns for girls under 15 years old will also use one-dose schedule.
Scenario B: One-dose schedules and gender-neutral vaccination	As in Scenario A. One-dose schedule for girls under 15 years old. LICs and LMICs adopt the one-dose schedule in 2023, while UMICs and HICs (including China) have a delayed switch in 2025.	As in Scenario A, but all countries adopt gender-neutral vaccination in the same year as the implementation of the one-dose schedule (unless already implemented).
Scenario C: Gender-neutral vaccination in HICs and MICs	As in Base Case. Two-dose schedule for girls under 15 years old. All countries introduce by 2028.	As in base case and all HICs and UMICs (including China) implement gender-neutral vaccination. Gavi-supported countries introduce with MACs for girls under 15 years old.
Scenario D: To achieve global cervical cancer elimination goals	As in base case, but assumes that all countries will achieve at least 90% coverage by 2030.	No MACs or vaccination of boys.

HIC: high-income country; LIC: low-income country; LMIC: lower-middle-income country; MIC: middle-income country; UMIC: upper-middle-income country.

**Table 2 vaccines-12-00004-t002:** HPV vaccines as of September 2022.

Manufacturer, Product	Subtype	Indicated for	Status
GlaxoSmithKline (GSK), Rixensart, Belgium Cervarix^®^	Bivalent	Persons 9–45 years of age	Licensed and prequalified
Xiamen Innovax Biotech subsidiary of Beijing Wantai, Xiamen, Fujian, China Cecolin^®^	Bivalent	Females 9–45 years of age	Licensed and prequalified
Shanghai Zerun, subsidiary of Yunnan Walvax Biotechnology, Shanghai, China Walrinvax^®^/Wo Ze Hui^®^	Bivalent	Females 9–30 years of age	Licensed in China
Merck Sharpe & Dohme (MSD), Rahway, NJ, USA Gardasil^®^	Quadrivalent	Persons 9–45 years of age	Licensed and prequalified
Serum Institute of India, Pune, India CERVAVAC qHPV^®^	Quadrivalent	Persons 9–26 years of age	Licensed in India
Merck Sharpe & Dohme, Rahway, NJ, USA Gardasil 9^®^	Nonavalent	Persons 9–45 years of age	Licensed and prequalified
China National Biotec Group, Beijing, China	Quadrivalent	To be confirmed	Phase 3 clinical trial
Nanolek, Moscow, Russian Federation	Quadrivalent	To be confirmed	Phase 3 clinical trial
Beijing Health Guard, Beijing, China	Quadrivalent	females 20–45	Phase 3 clinical trial
Bowei Biologics, Tianjin, China	Quadrivalent	females 9–45	Phase 3 clinical trial
Shanghai Zerun, subsidiary of Yunnan Walvax Biotechnology, Shanghai, China	Nonavalent	females 9–45	Phase 3 clinical trial
Xiamen Innovax Biotech subsidiary of Beijing Wantai, Xiamen, Fujian, China NCT04537156	Nonavalent	females 18–45	Phase 3 clinical trial
Jangsu Recbio Technology, Taizhou, Jiangsu, China REC603	Nonavalent	females 9–45	Phase 3 clinical trial

Bivalent: includes HPV subtypes 16 and 18. Quadrivalent: includes subtypes 6, 11, 16, and 18. Nonavalent: includes subtypes 6, 11, 16, 18, 31, 33, 45, 52, and 58.

## Data Availability

The demand and pricing data presented in this study are available on request from the corresponding author. The supply data are not publicly available due to 3rd Party Data Restrictions.

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
