# Peer review of "The Global Demand and Supply Balance of the Human Papillomavirus Vaccine: Implications for the Global Strategy for the Elimination of Cervical Cancer"

_vaccines, 2023, doi:10.3390/vaccines12010004_

Round 1
Reviewer 1 Report
Comments and Suggestions for Authors
The title of the paper is “The global demand and supply balance of the Human Papilloma virus vaccine; implications for the Global strategy for the elimination of cervical cancer” while the paper only discussed about the demand, supply and pricing of the HPV vaccine. There is nothing about the Elimination of cervical cancer.
What is meant by cervical cancer elimination? What are the targets for vaccine coverage? Targets for decrease in incidence and mortality? If there is any cervical elimination strategy already existing, please explain it.
This article contains valuable information, but I have doubts, if it falls in the category of original article or commentary or any other type of the article?
A typical abstract of an article is comprises of
Background
Methods
Results
Conclusion
Which are missing in the paper.
In discussion, discuss your results with the latest published data.
Author Response
Thanks a lot for the comments and the very useful suggestion to improve the quality of the paper. Here below please find our responses and the references to the updates implemented in the paper.
The title of the paper is “The global demand and supply balance of the Human Papilloma virus vaccine; implications for the Global strategy for the elimination of cervical cancer” while the paper only discussed about the demand, supply and pricing of the HPV vaccine. There is nothing about the Elimination of cervical cancer. What is meant by cervical cancer elimination? What are the targets for vaccine coverage? Targets for decrease in incidence and mortality? If there is any cervical elimination strategy already existing, please explain it.
More detailed reference to the Global strategy has been added from line 41 to 51 with clear reference to how the vaccination coverage contributes to the goal.
A typical abstract of an article is comprises of: Background, Methods, Results, Conclusion. Which are missing in the paper.
The abstract has been restructured as recommended for improved clarity.
In discussion, discuss your results with the latest published data.
The latest information on supply (table 2) and historical demand (line 158 thru 164) has been included presented in the results serve as the base for the discussion. The forecast remains unchanged as developed as part of the MI4A process during 2022.
Reviewer 2 Report
Comments and Suggestions for Authors
Even though this manuscript does not bring experimental data, I think it is of outmost interest for the readers of the Vaccine journal as it deals with a theme of general interest. It is also an ambitious project given the huge differences between countries.
The manuscript is well written and I have only some minor observations/questions.
Abstract, line 26: even though in principle I agree with the statement, I think it needs to be toned down just a little bit, in the sense that unexpected (even catastrophic events) should be taken into account.
Line 33: I am not sure I understood why the adoption of a 1 dose schedule would require “careful management” to ensure supply sustainability. Would you please care to explain this?
Introduction, Line 46: for the reader which is not familiar with this topic it would be useful to offer some details about Gavi.
Demand forecasts: just as a general comment, any such forecast should consider as well the vaccination refusal due to social media campaigns, misinformation, fake-news, conspiracy theories, etc.
Line 94: vaccination of girls of any age?
Line 97: could you please explain what you meant by “one through four of routine HPV vaccine use”?
Line 157: up to ten additional countries are planning introductions during 2023. Should we understand that no vaccinations were performed in these countries so far? We are very close now to 2024. Do you have any information regarding the situation in these 10 countries?
Line 173: again, do you happen to have any up-dated information about the situation in 2023?
Line 178: which are the Gavi-supported countries, let’s say in terms of income/capita?
Line 240: could you please comment on the manufacturers’ recommendations?
Line 242: this statement is somehow debatable as rich countries will probably be able to buy enough vaccines in order to cover more than one dose. This is actually an issue that you addressed in the discussion chapter
Author Response
Thanks a lot for all the valuable comments that help improving the quality of the manuscript.
Even though this manuscript does not bring experimental data, I think it is of outmost interest for the readers of the Vaccine journal as it deals with a theme of general interest. It is also an ambitious project given the huge differences between countries.
The manuscript is well written and I have only some minor observations/questions.
Abstract, line 26: even though in principle I agree with the statement, I think it needs to be toned down just a little bit, in the sense that unexpected (even catastrophic events) should be taken into account.
The text has been modified (line 24) to indicate that this is true under normal circumstances, hence that catastrophic events are not considered. Having a word limit in the abstract we could not further articulate.
Line 33: I am not sure I understood why the adoption of a 1 dose schedule would require “careful management” to ensure supply sustainability. Would you please care to explain this?
This referred to the risk of manufacturers withdrawing or reducing their supply in view of a reduced demand consequence of the adoption of a 1-dose schedule (by half for the demand already existing). The sentence in the abstract has been eliminated to comply with the word limit in the section. More extensive references can be found in the results (line 258 and following) section.
Introduction, Line 46: for the reader which is not familiar with this topic it would be useful to offer some details about Gavi.
A brief explanation of Gavi’s role has been added (line 40).
Demand forecasts: just as a general comment, any such forecast should consider as well the vaccination refusal due to social media campaigns, misinformation, fake-news, conspiracy theories, etc.
In the absence of specific data, we assumed the trend not to worsen compared to the current one already captured in the historical coverage data that serves as base for the future projections. The use of this coverage data to formulate future projections allows to maintain the dampened effect also in the future (line 91 and 190).
Line 94: vaccination of girls of any age?
This refers only to girls 10 to 14 years. This is now made explicit (line 99).
Line 97: could you please explain what you meant by “one through four of routine HPV vaccine use”?
Historically, coverage has followed a specific trend in most introducing countries - Year 1: 100%, Y2: 80%, Y3: 90%, Y4:100%. This dynamic has been used as the standard uptake for new introduction. The sentence was confusing, and it has been modified to provide clarity (line 103 -104).
Line 157: up to ten additional countries are planning introductions during 2023. Should we understand that no vaccinations were performed in these countries so far? We are very close now to 2024. Do you have any information regarding the situation in these 10 countries?
The sentence has been modified (line 163) providing full clarity on countries expected to introduce and that have effectively introduced. The total number of expected introductions has been also updated based on the latest intelligence.
Line 173: again, do you happen to have any up-dated information about the situation in 2023?
We do not have yet, data will be available only in summer 2024
Line 178: which are the Gavi-supported countries, let’s say in terms of income/capita?
Gavi supported countries are all LIC, and LMICs. The sentence has been modified for clarity (line 168).
Line 240: could you please comment on the manufacturers’ recommendations?
The rationale has been clarified in relation to potential price increases and market exits (line 270).
Line 242: this statement is somehow debatable as rich countries will probably be able to buy enough vaccines in order to cover more than one dose. This is actually an issue that you addressed in the discussion chapter.
Some HICs (UK, Ireland) have already switched to single dose. In this sentence we are just highlighting how lower demand for the main risk group (girls 9-14) resulting from a single dose schedule, can create an opportunity for vaccinating other risk groups while keeping the budget. unchanged We modified the sentence making the reference to the budget explicit, to make our point clearer (line 274).
Reviewer 3 Report
Comments and Suggestions for Authors
This work draws on the work of the WHO Market Information for Access initiative, and reports demand and supply forecasts for HPV vaccines, calculations of supply-demand balance, and trends in vaccine prices.
On the surface, the message that "In the long term, HPV supply will be more than sufficient to meet the goals of the Global Strategy for the elimination of cervical cancer" is encouraging, but I suggest that a more nuanced discussion of the possible effects of vaccine hesitancy, and strategies to address this, would be helpful. Is global coverage estimated for 2021 (line 156) an over-estimate?
It is good that some aspects of the impact of the COVID-19 pandemic on vaccine supply have been considered, including vaccine affordability (line 340) and declines in immunization coverage (lines 220-221 - this was a bit unclear). Have the innovations in vaccine development and regulation that occurred in relation to COVID-19 been applied to any of the HPV vaccines now in phase 3 clinical trials? What, if any, are the effects of ongoing COVID vaccine development, and in the same jurisdictions, influenza and RSV vaccine development? In Africa and perhaps elsewhere, are there competitive effects of demand to vaccinate against HPV and malaria?
Equity between countries/regions - Mention is made of LIC and MIC carrying a disproportionate share of HPV disease burden (lines 333-334). It would add to analysis to see the anticipated demand-supply balance by different categories of countries, as alluded to later in that paragraph. It might be helpful to consider in relation to regional trends reported in another paper from the Global Burden of Disease Group (Zhang, X., Zeng, Q., Cai, W. et al. Trends of cervical cancer at global, regional, and national level: data from the Global Burden of Disease study 2019. BMC Public Health 2021;21:894). What about interface with screening for cervical cancer in HIC (and maybe some MIC), where there may be equity issues within countries?
There is some evidence of international variation in the distribution of high-risk HPV types, and the possibility that this might influence vaccine effectiveness (Mammas IN, Vageli D, Spandidos DA. Geographic variations of human papilloma virus infection and their possible impact on the effectiveness of the vaccination programme. Oncol Rep. 2008;20(1):141-5). I understand that such evidence is weak, and indeed cross-protection may be adequate, but I think some comment needed in view of experience with VOC in the COVID-19 pandemic.
Author Response
Thanks a lot for your comments that help improving the quality of the manuscript.
This work draws on the work of the WHO Market Information for Access initiative, and reports demand and supply forecasts for HPV vaccines, calculations of supply-demand balance, and trends in vaccine prices.
On the surface, the message that "In the long term, HPV supply will be more than sufficient to meet the goals of the Global Strategy for the elimination of cervical cancer" is encouraging, but I suggest that a more nuanced discussion of the possible effects of vaccine hesitancy, and strategies to address this, would be helpful. Is global coverage estimated for 2021 (line 156) an over-estimate?
In the absence of data, we do not feel comfortable making a specific scenario. We have clarified that the impact of vaccine hesitancy is included as captured by the current historical coverage data that serves as the base for future projections (line 91 and 190). In this way, the vaccine hesitancy impact is considered unchanged over time. Stronger hesitancy will reduce demand further eliminating the risk of shortages.
It is good that some aspects of the impact of the COVID-19 pandemic on vaccine supply have been considered, including vaccine affordability (line 340) and declines in immunization coverage (lines 220-221 - this was a bit unclear).
The sentence has been modified to provide more clarity (line 232).
Have the innovations in vaccine development and regulation that occurred in relation to COVID-19 been applied to any of the HPV vaccines now in phase 3 clinical trials?
A similar acceleration has not materialized for HPV. The absence of the pressure of the pandemic has moved vaccines development back towards the pre-pandemic “normal” state. Furthermore, supply is now sufficient and the pipeline rather rich, reducing the need for the extra efforts required for a faster clinical development.
What, if any, are the effects of ongoing COVID vaccine development, and in the same jurisdictions, influenza and RSV vaccine development? In Africa and perhaps elsewhere, are there competitive effects of demand to vaccinate against HPV and malaria?
No effect has been recorded so far. With a more widespread adoption of the malaria vaccine, attention will be required in case impact of demand were to materialize.
Equity between countries/regions - Mention is made of LIC and MIC carrying a disproportionate share of HPV disease burden (lines 333-334). It would add to analysis to see the anticipated demand-supply balance by different categories of countries, as alluded to later in that paragraph. It might be helpful to consider in relation to regional trends reported in another paper from the Global Burden of Disease Group (Zhang, X., Zeng, Q., Cai, W. et al. Trends of cervical cancer at global, regional, and national level: data from the Global Burden of Disease study 2019. BMC Public Health 2021;21:894).
HPV caccine supply is global with all manufacturers seeking prequalification and targeting access to the global market. While a segmentation exists driven by commercial strategies (some manufacturers may target primarily certain markets – e.g., high price HICs or domestic ones) a complete separation of market segments does not exist that calls for definition of separate balances. The point has been clarified in the text (line 211 and following).
What about interface with screening for cervical cancer in HIC (and maybe some MIC), where there may be equity issues within countries?
Those are both components of the global strategy. This is now explicitly mentioned (line 46 and following). No impact is assumed on vaccination policies. HICs have all already functioning HPV vaccination programs so better screening does not seem to reduce the drive towards vaccination from a policy or vaccinee standpoint.
There is some evidence of international variation in the distribution of high-risk HPV types, and the possibility that this might influence vaccine effectiveness (Mammas IN, Vageli D, Spandidos DA. Geographic variations of human papilloma virus infection and their possible impact on the effectiveness of the vaccination programme. Oncol Rep. 2008;20(1):141-5). I understand that such evidence is weak, and indeed cross-protection may be adequate, but I think some comment needed in view of experience with VOC in the COVID-19 pandemic.
In view of the limited evidence, we are not considering any impact of VOC. Nonetheless the trend towards higher valency vaccines (nonavalents vs. quadri and bivalent) may, at least partially, provide better protection at least towards changes in the importance of existing strains and potentially provide additional cross-protective protection. A reference to this trend in vaccine development is now made explicit (line 206 and 247).
Round 2
Reviewer 3 Report
Comments and Suggestions for Authors
I appreciate the responses. No further comments